# Liquid chromatography coupled to high-resolution mass spectrometry metabolomics: A useful tool for investigating tumor secretome based on a three-dimensional co-culture model

Andrea C. Pelosi[1◉‡], Anna Maria A. P. Fernandes[1,2◉‡], Leonardo F. Maciel[3], Alex A. R. Silva[2], Giulia C. Mendes[3], Luísa F. Bueno[3], Lívia Maria F. Silva[3], Rafael F. Bredariol[3], Maycon G. Santana[4], Andreia M. Porcari[2], Denise G. Priolli[1] *

1 Health Sciences Postgraduate Program, São Francisco University, Bragança Paulista, São Paulo, Brazil, 2 MS4Life Laboratory of Mass Spectrometry, Health Sciences Postgraduate Program, São Francisco University, Bragança Paulista, São Paulo, Brazil, 3 Multidisciplinary Laboratory, Medical School, Sao Francisco University, Bragança Paulista, São Paulo, Brazil, 4 Multiprofessional Nursing Residency Program in Oncology, A.C. Camargo Cancer Center, São Paulo, Brazil

◉ These authors contributed equally to this work.
‡ ACP and AMAPF are share first authorship on this work.
* deprioli@gmail.com

**Data Availability Statement:** All metabolomics data are available via Metabolights with identifier

## Abstract

Three-dimensional (3D) cell culture technologies, which more closely mimic the complex microenvironment of tissue, are being increasingly evaluated as a tool for the preclinical screening of clinically promising new molecules, and studying of tissue metabolism. Studies of metabolites released into the extracellular space (secretome) allow understanding the metabolic dynamics of tissues and changes caused by therapeutic interventions. Although quite advanced in the field of proteomics, studies on the secretome of low molecular weight metabolites (< 1500 Da) are still very scarce. We present an untargeted metabolomic protocol based on the hybrid technique of liquid chromatography coupled with high-resolution mass spectrometry for the analysis of low-molecular-weight metabolites released into the culture medium by 3D cultures and co-culture (secretome model). For that we analyzed HT-29 human colon carcinoma cells and 3T3-L1 preadipocytes in 3D-monoculture and 3D-co-culture. The putative identification of the metabolites indicated a sort of metabolites, among them arachidonic acid, glyceric acid, docosapentaenoic acid and beta-Alanine which are related to cancer and obesity. This protocol represents a possibility to list metabolites released in the extracellular environment in a comprehensive and untargeted manner, opening the way for the generation of metabolic hypotheses that will certainly contribute to the understanding of tissue metabolism, tissue-tissue interactions, and metabolic responses to the most varied interventions. Moreover, it brings the potential to determine novel pathways and accurately identify biomarkers in cancer and other diseases. The metabolites indicated in our study have a close relationship with the tumor microenvironment in accordance with the literature review.

MTBLS5375 and http://www.ebi.ac.uk/metabolights/MTBLS5375.

**Funding:** A. A. R. S. is supported by the Coordination for the Improvement of Higher Education Personnel (CAPES, www.capes.gov.br, grant #88887.511153/2020-00). A. M. A. P. F. is supported by the Coordination for the Improvement of Higher Education Personnel (CAPES, www.capes.gov.br - grant #001). A. M. P. is supported by São Paulo Research Foundation (FAPESP, https://fapesp.br/, Grant #2019/04314-6). M. G. S. is supported by São Paulo Research Foundation (FAPESP, https://fapesp.br/, Grant #18/21906-1). D. G. P. is supported by São Paulo Research Foundation (FAPESP, https://fapesp.br/, Grant #18/21471-5 and #19/23592-7). No. The funders had and will not have a role in study design, data collection and analysis, decision to publish, or preparation of the manuscript.

**Competing interests:** The authors have declared that no competing interests exist.

## Introduction

Colorectal cancer (CRC) is the third most common cancer in the world and the second most deadly. Every year 1,8 million people are diagnosed and about 900,000 patients die from CRC [1]. It is usually diagnosed at advanced stages due to the limitations of current screening methods used in the clinic [2–4]. Only two blood-based biomarkers are available to monitor CRC patients: carcinoembryonic antigen (CEA) and carbohydrate antigen 19–9 (CA19-9). CEA, is a high molecular weight glycoprotein, found in embryonic tissue and colorectal malignancies. However, high levels of this compound in the blood are not specific for CRC and elevated levels of CEA are found in advanced stages of a fraction of CRC patients. The CA19-9 antigen, compared to CEA, is less sensitive and specific for CRC [5]. There is an urgent need to develop new biomarkers and modalities to detect, diagnose, and monitor the disease.

Pre-clinical *in vitro* evaluation is traditionally carried out in two-dimensional (2D) cell monoculture representing an easy and well-established methodology. The growth in 2D surface results in cell and cytoskeleton's flattening and remodeling, changing important factors in the tumor microenvironment *in vivo* such as nuclear form, protein and lipid synthesis, biochemical responses, and signaling cascades. It is widely held that 2D culture is unable to simulate the original tumor microenvironment, which grows three-dimensionally (3D) [6, 7]. This is why many compounds and drugs are active in 2D culture models but are not successful in subsequent preclinical tests [8, 9].

The 3D culture systems have received attention to avoid certain disadvantages of 2D-culture models [10, 11]. Three-D spheroids are formed by cell aggregation mediated by the interaction between integrin and extracellular matrix with subsequent compaction by transmembrane protein interactions such as E-cadherin [12, 13]. This allows three-dimensional cell cultures to structure similar to natural tissues, to present intercellular interactions and adhesions, and simulate *in vivo* tumor characteristics such as hypoxia, necrosis, invasion, metastasis, anti-apoptosis and drug resistance [14–18].

Metabolomics, an approach targeted at comprehensive profiling of the metabolites in a biological system, has demonstrated its great potential for use in the early diagnosis and personalized treatment of various cancers including CRC [19, 20]. By applying advanced analytical techniques and bioinformatics tools, the metabolome can be mined for biomarkers associated with carcinogenesis and prognosis [4].

The metabolome is the set of molecules below 1.5 kDa produced in response to intrinsic biological and environmental factors [21]. It is the net result of the integration of systemic metabolic processes and reveals the metabolite-enzyme relationships that regulate these processes. Thus, the metabolome, in contrast to genome or proteome, has been considered a more instantaneous representation of the phenotype, since its changes occur more quickly than changes in genes or proteins, and may indicate, in a time closer to the real one, current biological events. Additionally, several drug target a specific metabolite by inhibiting its enzyme or receptor, which are proteins and not genetic sequences. In this context, metabolomics studies seem more promising for discovering new molecules and metabolic pathways for potential therapeutic targets. Thus, recognizing that metabolites play significant and dynamic roles in biological processes has made metabolomics a key area in studies of systemic profiles in diseases and medicine in general [22].

Metabolomic studies that use liquid chromatography coupled to mass spectrometry (LC-MS) as an analytical technique, provide a comprehensive analysis of the metabolome and revolutionize the study of small molecules. LC-MS-based metabolomics can be categorized into (i) targeted analysis which is a pre-established quantitative analytical approach to a list of known metabolites and (ii) untargeted metabolomics which is characterized by the

simultaneous measurement of a large number of metabolites of each sample, usually without prior knowledge of the constituents and changes in them. The main advantage of untargeted metabolomics is the discovery of new metabolites in relation to the study context; therefore, it is considered a hypothesis-generating approach [23].

Two approaches are considered when performing metabolomic studies of cultured cell lines. Those focused on intracellular metabolites of isolated cells and those focused on the secretome or extracellular metabolites released (ERM) by cells in the culture medium. The analysis of ERM provides a picture of the metabolites resulting from the exchange carried out between the cells and the culture medium. This approach has the following advantages: ensuring little (or non-existent) handling of cells, which avoids the production of artifacts, allowing the monitoring of metabolic activity in response to experimental disturbances without cell disruption, enabling the monitoring of metabolic changes over time within the same culture and avoid carrying out long and multiple extraction procedures, which also enable the production of technical artifacts that can lead to concealment or unwanted manipulation of biological results [24]. However, the disadvantages of working with the secretome include matrix effects related to the salty media composition, which is rich in sugars, lipids, proteins, and water and might interfere in the analysis. The dilution of the metabolites of interest in the media also impacts the detection sensitivity [25].

Here we present a liquid chromatography coupled with high-resolution mass spectrometry metabolomic based protocol for the analyses of ERM. For the development of the protocol, culture medium from monoculture spheroids and co-culture (from HT-29 and 3T3-L1 cells) were analyzed. The putative identification of relevant molecular features for each spheroid type and those influenced by co-culture demonstrate the applicability of the method to the study of the metabolism of these spheroids and their tissue-tissue interactions with a focus on discovering new therapeutic targets, biomarkers and their associated metabolic pathways.

## Material and methods

The protocol described in this peer-reviewed article is published on protocols.io, [doi.org/10.17504/protocols.io.b24vqgw6] and is included for printing as S1 File with this article.

## Expected results

For this protocol, we used untargeted ultra-performance liquid chromatography coupled to electrospray ionization quadrupole time-of-flight mass spectrometry (UPLC-QTOF) operating in high energy collision spectral acquisition mode (MS$^E$) mode approach to investigate differences between extracellular metabolomic profiles of HT-29 and those of 3T3-L1 spheroids. The presence of adipose tissue can influence the development of CRC in vivo [26]. To better understand this influence in vitro, we used a model of HT-29 cell line as cancer cells and 3T3-L1 as adipocytes [27]. These cell lines were cultured separately and co-cultured, and their secretome was used to investigate the tumor microenvironment and the interaction between cancer cells and the adipocyte tissue.

CSH (charged surface hybrid) particles were designed to enable sample loading and improve peak symmetry when using low ionic strength mobile phases, as instructed by the manufacturer. Reversed-phase columns, such as the chosen CSH C18, are broadly used for metabolomics investigation in different matrices such as plasma [20], serum [28], urine [29], tissue [30], and cell cultures [31], as chromatographic columns of this type generally result in the detection of more features [32]. The ERM obtained after the co-cultivation of both cell types was investigated. In the beginning, a total of 2658 molecular features were detected in the positive ionization mode and 3521 features were detected in the negative ionization mode.

These features retain the information of retention time and mass-to-charge ratio (tR_*m/z*) of each metabolite elucidated after the LC-MS runs.

Volcano plot statistical analysis of all cultured conditions compared with the blank samples (cultured medium only) provide the number of characteristic metabolites of each spheroid as reported in Table 1.

Only molecular features with log2 (FC) > 0 were considered because they represent an increase of the metabolite abundance in the medium due to the contact with the organoids. The statistical comparison (volcano plot) of all three different conditions was also performed and returned some relevant metabolites of each cell type as depicted in Table 2. Although Table 2 brings a small number of metabolites when compared to the detected ones, these metabolites were the ones that met the statistical criteria of relevance, as well as the annotation criteria based on MS/MS and isotopic pattern recognition. Indeed, in untargeted metabolomics, linking chemical structures to the data obtained by mass spectrometry remains a significant challenge. The vast majority of information collected by metabolomics is the so-called "dark matter," i.e., chemical signatures that remain uncharacterized [33].

Arachidonic acid (ARA), 5-HETE, and dihomo-γ-linolenic acid (DGLA) stood out among the listed metabolites. Studies have related ARA in the colorectal cancer carcinogenesis process, with the influence of the inflammatory process on tumor growth and progression through the interaction of inflammatory cytokines and chemokines with tumor cells [34–36]. Cajal cells and F2d fibroblasts, mesenchymal components of colonic tissue related to CCR, have already demonstrated high concentrations of ARA metabolism genes [37, 38]. Therefore, ARA was studied as a therapeutic target due to its direct involvement in the process of inflammation and carcinogenesis of colorectal cancer. Corroborating to these data, the use of ibuprofen and aspirin by adult patients reduced the risk of progressing to cancer for premalignant and advanced-stage lesions, as well as for recurrent adenomas [39, 40], suggesting that ARA inhibition may indeed play an important role in colorectal carcinogenesis. Fig 1 shows the increase of some metabolites, ARA included, which were positively or negatively impacted by the CRC spheroid in the presence of adipocytes. Furthermore, studies demonstrate the increase of 5-lipoxygenase (5-LOX), part of the ARA pathway, and its metabolite, 5-HETE, in tumor tissues of the prostate, pancreas, colon, stomach and cervix [41–46].

DGLA is related to linoleic acid metabolism, an unsaturated fatty acid found in omega-6, which is associated with increased tumor growth, size, and metastatic potential [47]. Diets rich in omega-6 would be related to pro-inflammatory effects in the body, which may predispose to CRC development in long-term exposure [47, 48]. Also noteworthy are γ-glutamyltyrosine and γ-glutamylisoleucine as glutathione metabolites, a potential biomarker of tumorigenesis, beta-alanine, an indicator of tumor protein metabolism reprogramming, and histidinal, a metabolite of histidine, which is involved in several biological responses related to tumor growth [49–52]. The biological findings point to relevant aspects of tumor metabolism and highlight the potential of the protocol described here as a valuable tool in the metabolism study of *in vitro* model CCR based on 3D culture and co-culture cells.

**Table 1. Molecular features of 3D cultured cells.**

| Cell Type | Negative Mode | Positive Mode |
|---|---|---|
| HT-29 | 191 features | 3 features |
| 3T3-L1 | 329 features | 267 features |
| HT-29 + 3T3-L1 | 129 features | 14 features |

a. characteristic metabolites of each spheroid after comparison with the blank samples (cultured media only).

**Table 2. Representative secretome from 3D culture or 3D co-cultured spheroids.**

| Feature Code[a] | Log2 (FC)[b] | Putative assignment | Identifiers[c] | | Comparative abundances[d] | | |
|---|---|---|---|---|---|---|---|
| tR_m/z | | | | | HT-29 | 3T3-L1 | HT-29 + 3T3-L1 |
| 8.49_303.2320m/z | 1.5 | Arachidonic Acid | C00219 | HMDB0001043 | ✓ | NA | DOWN |
| 0.75_151.0247m/z | 3.2 | Glyceric Acid | C00258 | HMDB0000139 | ✓ | DOWN | DOWN |
| 8.59_329.2473m/z | 1.1 | Docosapentaenoic Acid* | C16513 | HMDB0001976 | ✓ | DOWN | DOWN |
| 0.61_134.0460m/z | -4.0 | Beta-Alanine | C00099 | HMDB0000056 | ✓ | DOWN | UP |
| 1.67_291.0973m/z | 9.3 | γ-Glutamyltyrosine | C03363 | HMDB0011741 | ✓ | NA | NS |
| 3.52_241.1184m/z | 6.8 | γ-Glutamylisoleucine | C03363 | HMDB0011170 | ✓ | NA | NS |
| 8.13_301.2162m/z | 3.5 | 5-HETE | C04805 | HMDB0011134 | ✓ | NA | NS |
| 9.00_305.2475m/z | 2.5 | Dihomo-γ-linolenic acid | C03242 | HMDB0002925 | ✓ | NA | NS |
| 0.77_153.0402m/z | 4.8 | Xanthine | C00385 | HMDB0000292 | ✓ | NA | NS |
| 0.54_139.0743n | 1.5 | Histidinal | C01929 | HMDB0012234 | NA | NA | ✓ |
| 0.56_251.1008n | 2.6 | Deoxyadenosine | C00559 | HMDB0000101 | NA | NA | ✓ |
| 4.58_245.0920m/z | 15.5 | Formyl-N-acetyl-5-methoxykynurenamine | C05642 | HMDB0004259 | NA | ✓ | NS |
| 0.54_802.6697m/z | 11.5 | PC(o-38:0) | C00958 | HMDB0013408 | NA | ✓ | NS |
| 0.61_232.0824m/z | 8.4 | 2-Keto-6-acetamidocaproate | C05548 | HMDB0012150 | NA | ✓ | NS |
| 4.43_407.1214m/z | 7.9 | 2-S-glutathionyl acetate | C14862 | HMDB0062198 | NA | ✓ | NS |
| 1.39_298.0970m/z | 7.7 | 5'-Methylthioadenosine | C00170 | HMDB0001173 | NA | ✓ | NS |
| 0.65_152.0566m/z | 7.6 | Guanine | C00242 | HMDB0000132 | NA | ✓ | NS |
| 0.50_364.2445m/z | 7.0 | MAG(14:1) | C01885 | HMDB0011531 | NA | ✓ | NS |

a. tR = retention time; m/z = mass-to-charge ratio.

b. Compared to blank samples (culture media only).

c. HMDBXXXXXXX, metabolites described in the Human Metabolome Database (HMDB—https://hmdb.ca/); CXXXXX, described in the Kyoto Encyclopedia of Genes and Genomes database (KEGG—https://www.genome.jp/kegg/).

d. ✓ = presence; NA = absence; UP and DOWN = more or less abundant respectively, when compared with HT-29 culture; NS = not significantly impacted when compared with HT-29 culture.

* Annotated by exact mass only.

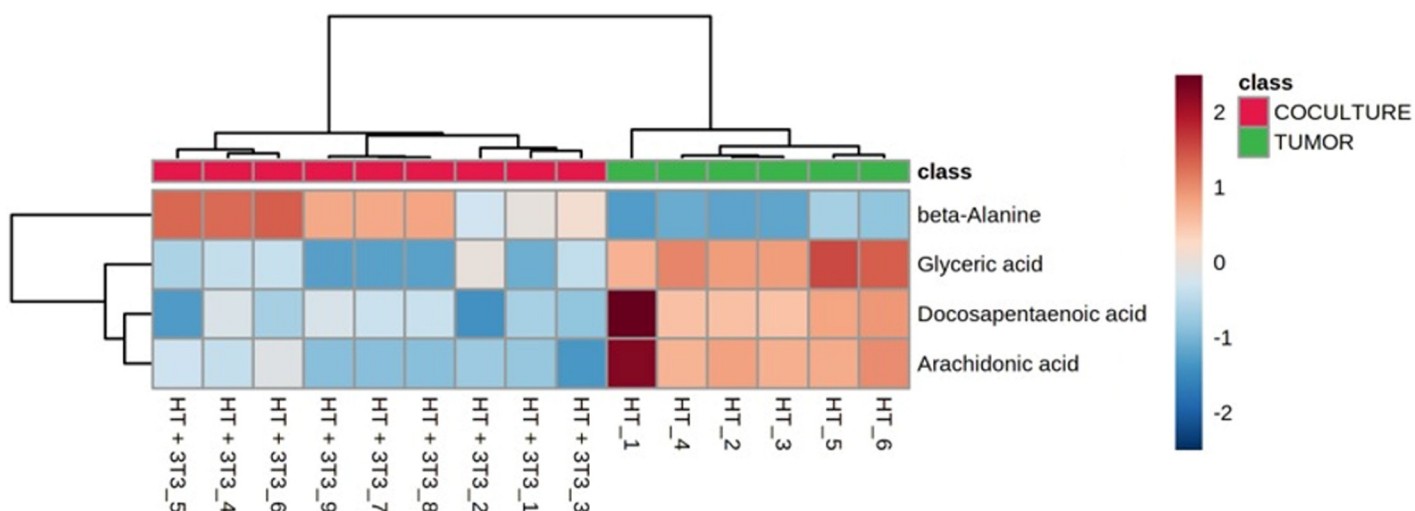

**Fig 1. Heatmap of the metabolites from HT spheroids impacted by the co-culture with 3T3 adipocytes.** The colors are normalized to the relative abundance of each metabolite. Samples (vertical axis) and metabolites (horizontal axis) are separated by Ward's algorithm and the dendrogram was scaled using Pearson's correlation. The clusters containing tumor spheroid alone and co-culture with tumor and adipocyte spheroids are highlighted in green and red, respectively.

## Supporting information

**S1 File. Protocol to secretome investigation of tumor 3D co-culture model.**
(PDF)

## Author Contributions

**Conceptualization:** Andrea C. Pelosi, Denise G. Priolli.

**Data curation:** Anna Maria A. P. Fernandes, Alex A. R. Silva.

**Formal analysis:** Andrea C. Pelosi, Anna Maria A. P. Fernandes, Leonardo F. Maciel, Alex A. R. Silva, Denise G. Priolli.

**Funding acquisition:** Anna Maria A. P. Fernandes, Alex A. R. Silva, Maycon G. Santana, Andreia M. Porcari, Denise G. Priolli.

**Investigation:** Leonardo F. Maciel, Luísa F. Bueno, Lívia Maria F. Silva, Rafael F. Bredariol.

**Methodology:** Andrea C. Pelosi, Anna Maria A. P. Fernandes, Alex A. R. Silva, Giulia C. Mendes, Maycon G. Santana.

**Project administration:** Andreia M. Porcari, Denise G. Priolli.

**Supervision:** Denise G. Priolli.

**Visualization:** Andrea C. Pelosi, Leonardo F. Maciel.

**Writing – original draft:** Andrea C. Pelosi, Anna Maria A. P. Fernandes.

**Writing – review & editing:** Andreia M. Porcari, Denise G. Priolli.

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
