## [Decision Letter · Decision Letter 0]

12 Jul 2022

PONE-D-22-05896High resolution liquid chromatography coupled to high resolution mass spectrometry metabolomics: A useful tool for investigating tumor secretome based on a three-dimensional co-culture modelPLOS ONE

Dear Dr. Denise Gonçalves Priolli,

Thank you for submitting your manuscript to PLOS ONE. After careful consideration, we feel that it has merit but does not fully meet PLOS ONE’s publication criteria as it currently stands. Therefore, we invite you to submit a revised version of the manuscript that addresses the points raised during the review process.

We look forward to receiving your revised manuscript.

Kind regards,

Tommaso Lomonaco, Ph.D

Academic Editor

PLOS ONE

Journal Requirements:

Alex A. R. Silva is supported by the Coordination for the Improvement of Higher Education Personnel (CAPES, #88887.511153/2020-00). Anna Maria A. P. Fernandes is supported by the Coordination for the Improvement of Higher Education Personnel (CAPES - grant #001 to A.M.A.P.F.). Andreia de Melo Porcari is supported by São Paulo Research Foundation (FAPESP) Grant (#2019/04314-6). Maycon Giovani Santana is supported by São Paulo Research Foundation (FAPESP) Grant (#18/21906-1). Denise Gonçalves Priolli is supported by São Paulo Research Foundation (FAPESP) Grant (#18/21471-5) and (#19/23592-7). Andrea Corazzi Pelosi has a scholarship from Sao Francisco University.

However, funding information should not appear in the Acknowledgments section or other areas of your manuscript. We will only publish funding information present in the Funding Statement section of the online submission form. 

A. A. R. S. is supported by the Coordination for the Improvement of Higher Education Personnel (CAPES, www.capes.gov.br, grant #88887.511153/2020-00). A. M. A. P. F. is supported by the Coordination for the Improvement of Higher Education Personnel (CAPES, www.capes.gov.br - grant #001). A. M. P. is supported by São Paulo Research Foundation (FAPESP, https://fapesp.br/, Grant #2019/04314-6). M. G. S. is supported by São Paulo Research Foundation (FAPESP, https://fapesp.br/, Grant #18/21906-1). D. G. P. is supported by São Paulo Research Foundation (FAPESP, https://fapesp.br/, Grant #18/21471-5 and #19/23592-7).

No. The funders had and will not have a role in study design, data collection and analysis, decision to publish, or preparation of the manuscript.

Reviewers' comments:

Reviewer's Responses to Questions

**Comments to the Author**

1. Does the manuscript report a protocol which is of utility to the research community and adds value to the published literature?

Reviewer #1: Yes

2. Has the protocol been described in sufficient detail?

Descriptions of methods and reagents contained in the step-by-step protocol should be reported in sufficient detail for another researcher to reproduce all experiments and analyses. The protocol should describe the appropriate controls, sample sizes and replication needed to ensure that the data are robust and reproducible.

Reviewer #1: Partly

3. Does the protocol describe a validated method?

Reviewer #1: Yes

4. If the manuscript contains new data, have the authors made this data fully available?

Reviewer #1: No

**5. Is the article presented in an intelligible fashion and written in standard English?**

Reviewer #1: Yes

6. Review Comments to the Author

Reviewer #1: The manuscript by Pelosi et al. describes a protocol for 3D cell culture and its secretome analysis by liquid chromatography-mass spectrometry based-metabolomics. Overall, the manuscript is easy to follow and provides a protocol on a recent field of study. There is a growing interest in 3D cell culture metabolomics in the literature. However, the LC-MS based-metabolomics section of the protocol is poorly described, missing various important information. Because of this and other issues pointed out below, I recommend a revision of the manuscript.

Title

Please, use “liquid chromatography coupled to high-resolution mass spectrometry” instead of “High resolution liquid chromatography coupled to high resolution mass spectrometry” (also check other manuscript sections since there is a lack of formatting). I cannot see any reason to use “high resolution liquid chromatography” since metabolomics applications are mainly performed using columns with sub-2 um particles.

Abstract

Correct “high-performance liquid-chromatography coupled with high resolution mass spectrometry” to “liquid chromatography coupled to high-resolution mass spectrometry”

Introduction

The advantages of using the secretome instead of intracellular metabolites of isolated cells for metabolomics are stated. But what are the disadvantages/limitations of this approach? The authors need to provide some thoughts on it.

Materials and methods (protocols.io)

Step 20: specify how the culture medium is collected from the plate. And please, confirm if it is 50 mg of IPA or if it should be 50 uL.

Step 21: use G force (SI unit) instead of RPM.

Step 23: specify which “fluoride-phenylalanine” standard was used, its source/catalog number, and solvent.

Step 24: confirm if the formic acid was indeed not added to acetonitrile.

Step 27: what are the reasons for such low column over temperature (20 °C)?

Step 34: detail the parameters used for RAW data processing in Progenesis IQ software.

Step 36: what were the MS/MS and isotopic similarity scores employed?

Step 37: detail how data was pre-processed (scaling, normalization, etc).

Information about quality controls (pooled QC, standard mix solution) and blanks (extraction blanks, solvent blanks) are missing.

Information about the number of samples investigated and the replicates are missing.

Expected results

It is unclear why the authors have chosen to co-culture HT-29 human colon carcinoma cells with 3T3-L1 preadipocytes. An explanation is missing in the manuscript.

Please, inform in Table 1 that the molecular features refer to comparison with a blank sample

An explanation about why choosing the CSH C18 column to perform the secretome analysis must be provided since column chemistry greatly impacts the class of compounds detected.

There is a considerable discrepancy between the number of features detected and metabolites annotated in Table 2. So, a brief discussion should be provided about it.

Why is there “m/z” after the feature code (8.49_303.2320m/z) in table 2?

The correct representation of retention time is tR (R subscripted) and not RT.

Raw data files should be made available on data repositories (e.g., Metabolomics Workbench and MetaboLights).

7. PLOS authors have the option to publish the peer review history of their article (what does this mean?). If published, this will include your full peer review and any attached files.

Reviewer #1: No

---

## [Author Response · Author response to Decision Letter 0]

23 Aug 2022

The response letter with the itemized comments to the reviewer was uploaded as a separated file, named "response to reviewers"

---

## [Editor Report · Decision Letter 1]

1 Sep 2022

Liquid chromatography coupled to high-resolution mass spectrometry metabolomics: A useful tool for investigating tumor secretome based on a three-dimensional co-culture model

PONE-D-22-05896R1

Dear Dr. Priolli ,

We’re pleased to inform you that your manuscript has been judged scientifically suitable for publication and will be formally accepted for publication once it meets all outstanding technical requirements.

Kind regards,

Tommaso Lomonaco, Ph.D

Academic Editor

PLOS ONE

---

## [Editor Report · Acceptance letter]

12 Sep 2022

PONE-D-22-05896R1 

Liquid chromatography coupled to high-resolution mass spectrometry metabolomics: A useful tool for investigating tumor secretome based on a three-dimensional co-culture model 

Dear Dr. Priolli:

I'm pleased to inform you that your manuscript has been deemed suitable for publication in PLOS ONE. Congratulations! Your manuscript is now with our production department. 

Kind regards, 

on behalf of

Dr. Tommaso Lomonaco 

Academic Editor

PLOS ONE